# Self-Determination Research: Current and Future Directions

**DOI:** 10.3390/bs14070613

**Published:** 2024-07-19

**Authors:** Kathryn M. Burke, Karrie A. Shogren, Andrea Parente, Abdulaziz Alsaeed, Austin M. Myers, Shawn Aleong

**Affiliations:** 1Institute on Disabilities, Temple University, Philadelphia, PA 19122, USA; andrea.parente@temple.edu (A.P.); shawn.aleong@temple.edu (S.A.); 2Kansas University Center on Developmental Disabilities, University of Kansas, Lawrence, KS 66045, USA; shogren@ku.edu (K.A.S.); aziz@ku.edu (A.A.); amyers94@ku.edu (A.M.M.)

**Keywords:** self-determination, intellectual and developmental disabilities, secondary transition, research

## Abstract

This article summarizes the history, current status, and future directions of self-determination research across the globe, with a focus on applications to the education of students with intellectual and developmental disabilities and their transition from school to adult life. Research on the development, implementation, and outcomes of self-determination assessments and interventions is explored. Causal Agency Theory, a theoretical framework for understanding the development of self-determination as a psychological construct, is reviewed, along with research on the importance of self-determination for inclusion, psychological growth, and overall well-being. Specific approaches, models, and perspectives for addressing the support needs of students with intellectual and developmental disabilities, particularly during transitions, are discussed. Assessment and intervention aligned with Causal Agency Theory, including the Self-Determination Inventory and the Self-Determined Learning Model of Instruction, are introduced. Future directions and emerging areas of research are summarized, including issues related to cultural validity, integration of strengths-based approaches, emerging technologies, and systemic changes in schools and communities.

## 1. History of Self-Determination in Intellectual and Developmental Disabilities

Self-determination is defined as an individual determining their own fate or course of action (free will) or as the right of a group of people to determine their own political status [1]. In the individual sense, self-determination involves acting or causing things to happen in one’s life [2], and in the communal sense, self-determination refers to groups of people having agency over their lives and culture. Nirje [3] was the first to discuss self-determination in the field of disability, as a call to action regarding self-determination as a right of people with disabilities. Nirje described the historic denial of opportunities for people with disabilities to express and take action based on their “choices, wishes, desires and aspirations” [3] (p. 177), and he noted recent progress for people with disabilities “to gain meaningful control over their own destinies” [3] (p. 178), fueled by the power of advocates. These values continue to be reflected in policies and human rights treaties around the globe, including the United Nations Convention on the Rights of Persons with Disabilities (UNCRPD; [4]).

In special education, the study of self-determination gained traction in the 1990s in the U.S. through the voices of disability advocates [5] and with the backing of funding from the U.S. Department of Education, Office of Special Education Programs (OSEP; [6]). OSEP funded model demonstration projects to develop, test, and disseminate knowledge, practices, and resources to promote self-determination with transition-age youth, given data on the dismal postschool outcomes of students with disabilities in the areas of employment, postsecondary education, and community living [7]. Researchers noted the disparities in outcomes related to disability (particularly for students with complex support needs), gender, and race and ethnicity, and pointed to the need for transition planning to be responsive to students’ individual goals, strengths, and needs.

With the support of federal funding, researchers developed interventions to promote skills associated with self-determination such as problem solving [8], choice making [9], and self-advocacy [10], as well as definitional frameworks for self-determination as a construct within the field of special education (e.g., [11,12]). In the decades since, the field has continued to advance knowledge across the globe with validated self-determination assessments and instructional strategies in multiple languages and cultures with documented impacts on outcomes. Such work includes validated measures of self-determination for youth and adults with and without disabilities [13,14] and instructional frameworks and content-specific curricula (e.g., [15]) designed to support students’ overall self-determination and skills and abilities associated with self-determination (e.g., self-advocacy; [16]). There have also been analyses of the relationship between self-determination and students’ postschool outcomes (e.g., [17,18]) and quality of life [19]. In this paper, we describe advancements in the research on self-determination, the status of knowledge and related practice and policy, considerations for addressing the supports needs of transition-age youth with intellectual and developmental disabilities, and directions for the future.

## 2. Current Status of Self-Determination Research around the World

While self-determination has received considerable attention as an essential outcome for people with intellectual and developmental disabilities internationally, there has been less attention to its different operationalization across cultures and cultural identities. For instance, the value placed on individualism versus collectivism within a culture may influence the focus of a person’s self-determined action. As an example, in Turkey, parent–child relatedness is a relevant factor in a young person’s self-determination, affecting the goals they choose to pursue and the plans they make [20]. Globally, ongoing work has advanced understanding of the definition of self-determination, support for the development of self-determination, and interventions to promote equity in self-determination outcomes for all students, inclusive of students with intellectual and developmental disabilities.

### 2.1. Human and Disability Rights

The United Nations Convention on the Rights of Persons with Disabilities (UNCRPD), the most significant international treaty on the rights of people with disabilities with 164 signatory states [21], emphasizes self-determination through language around choice and autonomy [4]. Its stated purpose is “to promote, protect and ensure full and equal enjoyment of all human rights and fundamental freedoms by all persons with disabilities, and to promote respect for their inherent dignity” [4] (p. 4). The first principle of the UNCRPD is “respect for inherent dignity, individual autonomy including the freedom to make one’s own choices, and independence of persons” [4] (p. 5). The UNCRPD further outlines specific rights related to self-determination across life domains. For example, Article 19 discusses the rights of people with disabilities to live in the community and ensures that “persons with disabilities have the opportunity to choose their place of residence and where and with whom they live” [4] (p. 13). Article 12 requires states and jurisdictions to recognize the legal capacity of people with disabilities “on an equal basis with others in all aspects of life” [4] (p. 10). The UNCRPD has specified that such recognition requires the abolishment of substitute decision-making legal regimes, such as guardianship and conservatorship [22].

The task of monitoring the implementation and effects of the UNCRPD internationally remains complex, with studies emerging in recent years attempting to operationalize indicators for such monitoring [23,24]. The United States remains one of few countries which has not ratified the UNCRPD [21] but has developed policy in recent decades which might be viewed as furthering the spirit of self-determination present within the UNCRPD. For example, in line with Article 12’s promotion of legal capacity for people with disabilities, an increasing number of states are introducing supported decision-making agreements as an alternative to guardianship structures [25]. Like Article 19 of the UNCRPD, federal waiver funding in the U.S. requires eligible states to offer a range of options for home-and-community-based settings integrated into the community from which people with disabilities can chose [26]. There is a need for continued research on the alignment of different countries’ disability policy with the UNCRPD and its emphasis on self-determination.

### 2.2. Inclusive Research

Inclusive research, in which people from the communities of focus participate as members of the research team [27], connects closely with the application of self-determination to the field. Inclusive research allows people with disabilities to contribute to research across the planning, conducting, and disseminating phases. Through inclusive research, co-researchers act as causal agents for change in the research process [28]. It adds an additional layer of authenticity and allows the research team to tailor their findings to that specific community. Two strategies can add to the success of inclusive research on self-determination. First, involve people with disabilities and provide the opportunity for them to lead research efforts: “If you support advocates to be co-researchers, your research will be that much more believable. If you get input from people with disabilities, the research can end up being more true and maybe not just made up by people who do not have that perspective” [29]. Second, identify the research methods that work best for people with disabilities. Determine how the person will contribute their strengths: “Some people may want to be involved all the way through and become a researcher like me, but other people may really like communicating the results or developing infographics, or just participating. Someone else may be good at connecting with people in interviews or mentoring people as part of an intervention” [29].

Shawn Aleong is a student researcher and self-advocate who has engaged in research on voting rights and plain language. He provides the following perspective: “Inclusive research is so important because it is so important to tell the researchers how to conduct the research for their community, and it is also super important to have someone with lived experience in every aspect of the research”. Communications specialist and self-advocate Austin Myers adds: “Inclusive research allows for the research team to gain that perspective from people with lived experience. It allows for the research to seem more authentic. As one of my colleagues stated, it also allows for better outcomes for those specific communities. It also allows said communities to be able to trust the research more”. Myers has supported self-determination research dissemination efforts and contributes by creating visual representations of research findings. He focuses on accessible dissemination for people with disabilities, making sure the product is easily readable and can be easily understood. The degree to which people with disabilities are included as co-researchers is an area for future study, and future standards for high quality research and research policy should consider adding the meaningful involvement of people with disabilities to criteria [30], as funding organizations have begun to do [31].

### 2.3. Connections to Outcomes

A body of research supports the relationship between self-determination and a range of outcomes for individuals with disabilities. Self-determination is associated with positive in-school and postschool outcomes for adolescents with intellectual and developmental disabilities, including academic achievement [32] and quality-of-life outcomes [33]. In a study with 100 students with disabilities in the city of Catalonia, a Spanish autonomous community, Mumbardó-Adam et al. found self-determination had the highest influence on quality of life across all variables [33].

Positive outcomes are attributed not only to self-determination as a personal characteristic but also to systemic variables which contribute to opportunities for individuals with disabilities to exercise self-determination. Research shows people with disabilities have more opportunities to exercise choice over their lives when residing in more community-integrated settings, rather than segregated or congregate settings (e.g., [34,35]). Opportunities to exercise self-determination and the associated skills and abilities, in turn, are associated with positive outcomes. People with disabilities report a higher quality of life when receiving services through a consumer-directed service delivery model, rather than an agency-directed model [36]. Similarly, people with disabilities are more likely to report that their support services facilitate a vision of a good life when elements of person-centered planning are present, including staff competency in soliciting the preferences of the people they support [37]. People with disabilities who have choice over supports and services in areas such as home, employment, and healthcare are significantly more likely to have a higher quality of life in domains including safety, freedom from abuse and neglect, health, community integration, and personal relationships [38].

### 2.4. Disparities

While a substantial body of research demonstrates the relationship between self-determination for transition-age youth with intellectual and developmental disabilities and positive life outcomes (e.g., employment, community living, community participation), data on self-determination reflect significant disparities based on disability status and race and ethnicity (e.g., [39,40]). Such disparities include lower levels of self-determination for students with autism, intellectual disability, and multiple disabilities compared to peers with other disabilities in the U.S. [40]. The lack of consistent implementation of interventions to promote self-determination in schools may be associated with the variability in self-determination across demographic factors, and research on this topic is needed. Existing research across cultural contexts suggests the need for an ongoing focus on creating opportunities for inclusive supports, research, and self-determination across cultures [14,41,42].

Furthermore, systematic reviews of the literature have shown issues with the limited inclusion of students with diverse personal characteristics and a lack of information reported on participant demographics. To reach equitable outcomes for students across personal characteristics, students from diverse backgrounds must be included in self-determination research and have consistent access to evidence-based practices to promote self-determination implemented with fidelity in schools. Additionally, there is a need for ongoing research understanding the barriers to self-determination for transition-age youth with intellectual and developmental disabilities at a global level. Vaucher et al. explored self-determination for adults with intellectual and developmental disabilities in residential institutions in Switzerland, finding that environmental constraints in institutional settings (e.g., rigid schedules), social bias (e.g., few job opportunities), and family dynamics (e.g., being infantilized by relatives) were among a range of barriers to self-determined action [43]. Understanding barriers to self-determination is critical for both systemic changes to access and supports and targeted opportunities for students to learn how to overcome such barriers.

## 3. Current Theoretical Perspectives

Scholars have advanced significant theoretical work on self-determination and disability since its early conceptualizations in the 1990s. Wehmeyer introduced the functional model of self-determination, in which self-determination is defined as “the attitudes and abilities required to act as the primary causal agent in one’s life and to make choices regarding one’s actions free from undue external influence or interference” [44] (p. 305). Most recently, Shogren and colleagues [2] introduced Causal Agency Theory as an extension of and revision to the functional model of self-determination based on advances in the field, including the international research and inclusive research that has further informed understandings of self-determination.

### 3.1. Causal Agency Theory

Within Causal Agency Theory, self-determination is defined as a “dispositional characteristic manifested as acting as the causal agent in one’s life. Self-determined people (i.e., causal agents) act in service to freely chosen goals. Self-determined actions function to enable a person to be the causal agent in his or her life” [2] (p. 258). A primary reason cited in revisiting the functional model was the emergence of positive psychology. Martin Seligman coined the term “positive psychology” as he called for science to shift to understanding and supporting the positive qualities in individuals [45]. This changing approach is evident in strengths-based understandings of disability, another key reason for the introduction of Causal Agency Theory. In a strengths-based approach, the focus shifts from a deficit-based or medical model to an understanding of disability through a person’s support needs based upon the demands of an environment. This social-ecological model also includes personally defined quality-of-life outcomes as the goal of disability supports and services, within which self-determination is an associated domain [46].

#### Strengths and Limitations

Causal Agency Theory aligns with advancements in the understanding of self-determination as a psychological construct. A significant strength of the theory is the degree to which self-determined action, rather than behavior, is emphasized as the driver of causal agency [2]. Causal Agency Theory outlines the three essential characteristics of self-determination (volitional action, agentic action, and action–control beliefs, referred to as decide, act, and believe in plain language developed by co-researchers with intellectual and developmental disabilities) and the related skills and attitudes (making choices and expressing preferences, solving problems, engaging in making decisions, setting and attaining goals, self-managing and self-regulating action, self-advocating, and acquiring self-awareness and self-knowledge [47]). By distinguishing the key elements of how people become self-determined, the field is better able to develop assessment and intervention to support growth and related outcomes as well as consider how these approaches can be culturally valid for people across the world. Causal Agency Theory also further details the difference between self-caused action and control, a misunderstanding in the field noted by Wehmeyer [48] as particularly problematic when applied to promoting self-determination with youth and adults with complex support needs. Specifically, it is important to recognize that self-determination does not mean independence or absolute control but reflects interdependence and building systems of support that advance individual and communal goals and values.

Causal Agency Theory discusses the importance of socio-contextual influences and how this can shape how individuals respond to challenges in their environment with causal actions. Shogren et al. describe one of the factors in the reconceptualization of the functional model of self-determination as the contextual changes to special education, with more attention on “inclusive practices, access to the general education curriculum, and multi-tiered systems of supports” [2] (p. 255). Given this context, Causal Agency Theory provides a framing of self-determination for not only students with disabilities but for all students, recognizing the importance of advancing opportunities for inclusion in all areas of life across the globe, consistent with the UNCRPD and work to align implementation of the UNCRPD with advancing self-determination and quality of life.

More recent movements in the field underscore the critical effect of systemic and structural oppression caused by racism, ableism, classism, and sexism, among other forms of bias. These systemic and structural injustices create challenges a person may not be able to overcome in pursuit of their self-directed goals, even with support, until societal change occurs [49]. To address the need for cultural responsivity in self-determination research and practice, Stansberry Brusnahan and colleagues [50] developed an intersectional self-determination framework for a social-justice-informed approach to culturally and linguistically sustaining practices. While this framework offers a valuable approach to practice, future theoretical work may expand upon Causal Agency Theory to explain not only how people become self-determined but how systemic barriers influence this process. Researchers can then more deeply explore the types of interventions and supports that are effective in counteracting externally imposed limitations and lack of opportunities across the globe, particularly the wide range of oppressions that limit opportunities for people with disabilities around the world during transition [51].

## 4. Addressing the Support Needs of Students with Intellectual and Developmental Disabilities

The supports paradigm refers to the conceptualization of disability through an individual’s support needs rather than deficits, with a focus on designing and implementing human service systems to identify and address support needs and build systems of supports that promote human flourishing and self-determination [52,53]. Below, we provide the current perspectives and considerations for addressing the support needs of transition-age students with intellectual and developmental disabilities with regard to self-determination and in-school and postschool outcomes.

### 4.1. Social-Ecological Approach

The social-ecological model examines human behavior in terms of the interactions between people and their environment [54]. When applied to disability, the social-ecological model is a framework for understanding disability and the role of disability service systems [55]. It can be understood in contrast to the medical model, in which a professional’s role is to reduce or eliminate what is labeled in this model as areas of deficit to change a person with a disability [56]. Unlike the medical model, the social-ecological model approaches disability through an analysis of fit between the person and their environment. While this model does not deny that disability entails a person having support needs that may differ in type or intensity from those of peers without disabilities, it directs professional efforts to prioritize the modification of the environment to better fit the individual. The social-ecological model can be a valuable framework for promoting self-determination for individuals with intellectual and developmental disabilities [54].

The social-ecological model aligns with the framework for self-determination interventions, where the focus is on a strength-based approach to help build an individual’s capacity, emphasizing accommodations or modifications within the individual’s environment and in the systems that support them. In the context of intellectual disability, the social-ecological model places emphasis on understanding the supports a person needs to operate within their environment and directing professional efforts to secure needed supports [56]. Researchers working from a social-ecological model have developed assessment tools to identify individuals’ support needs, such as the Supports Intensity Scale—Adult Version (SIS-A; [57]). Support needs assessments have also been introduced into K-12 settings [55,58], including a children’s version of the SIS [59].

### 4.2. Tiered Systems

Multi-tiered systems of supports (MTSS) offer a comprehensive framework to implement school-wide interventions with scaled supports and instruction individualized to students’ needs [60]. In MTSS, students receive evidence-based, universally designed instruction with tiers of more intensive and/or targeted instruction and support based on individual needs. Models of tiered universal supports include school-wide positive-behavior interventions and supports (SWPBIS; [61]) and response to intervention (RTI; [62]). Researchers have shown the positive academic and behavioral outcomes of MTSS [63]; however, MTSS has not included students with complex support needs with intentionality and consistency across research and practice [64]. The field has called for research exploring interventions to promote self-determination from an MTSS approach for students with and without disabilities, including students with complex support needs. However, it is necessary that global efforts to promote access to inclusive education underlie efforts to advance MTSS, so students with complex support needs are included.

Tools and practices to measure and support the development of self-determination, such as the Self-Determination Inventory: Student Report [SDI:SR] and the Self-Determined Learning Model of Instruction [SDLMI], are well suited for the core components of MTSS—universal screening and triangulation [65] and tiered instruction and supports [60]. More work is needed to establish how to support educators to implement the SDLMI with all students, promoting the inclusion of students with disabilities at varying intensities, how to implement universal self-determination instruction, and how to link self-determination instruction to other MTSS frameworks to impact a range of outcomes, such as academic, behavior, and social-emotional skills and abilities.

### 4.3. Students with Complex Support Needs

The population of students with complex support needs includes students with autism, intellectual disability, or multiple disabilities who have ongoing pervasive support needs across life domains [66]. In a systematic review, Alsaeed et al. [67] identified 10 research studies on interventions to promote self-determination for students with complex support needs, with evidence supporting the positive outcomes for self-determination status and skills and abilities associated with self-determination (e.g., goal setting, planning). No studies in the review were conducted in inclusive general education or elementary settings, suggesting the need to extend research and practice to promote self-determination for students with complex support needs across K-12-inclusive education settings. Alsaeed et al. also raised concerns on the limited and inconsistent descriptions of participants and their support needs and training and intervention procedures. The findings from the research support the usability and effectiveness of the interventions to promote self-determination with this population, but it is challenging to make recommendations for practice given the limited information on individualized modifications and supports.

### 4.4. Life Course Perspectives

Self-determination develops in early childhood and can be supported throughout the life course as people access autonomy-supportive environments across home, school, and community contexts. Thus, a key consideration for researchers is how autonomy-supportive environments across home, school, and community contexts impact the development of self-determination at an early age. Specific skills and abilities associated with self-determination are particularly applicable for younger children, including choice making, problem solving, decision making, goal setting and attainment, and self-regulation [68]. However, there is limited work focused on younger students with disabilities; a review of the research on self-determination instruction to improve reading outcomes for students with or at risk for learning disabilities showed that the majority of the research has focused on the specific subcomponent of self-regulation [69]. Building self-determination abilities for younger students has been identified as a critical foundation for these skills in secondary school and throughout adulthood [70]. Research on practices to support self-determination across the life course is needed, particularly given students’ articulation of ways that self-determination influenced them from childhood. Research must also extend throughout adulthood, thinking beyond just the transition to adulthood but also identifying considerations for older adults [71].

### 4.5. Particular Needs during Transitions

Transitions offer important opportunities for people to make decisions about their lives, and the transition from high school to the adult world is a pivotal time for adolescents to choose where and how they will live, learn, work, and participate in their communities. An extensive research base supports the importance of self-determination intervention and supports during the transition-planning process (e.g., [72]), although more research is needed in elementary settings and throughout adulthood, as noted.

The transition to adulthood is a particularly important time to think about self-determination abilities. For example, in 2015, Rhode Island, a state within the U.S., implemented a state-wide initiative to improve transition outcomes for secondary students with intellectual disability. The initiative resulted from a Consent Decree with the U.S. Department of Justice due to “unnecessary over-reliance upon segregated sheltered workshops and facility-based day programs” as primary postsecondary transition outcomes for students with intellectual disability [73]. As part of the initiative to improve transition outcomes, teachers across the state received training and implemented the SDLMI and Whose Future Is It? [15], a transition-planning curriculum. Future work could include the study of beginning self-determination instruction and supports earlier, even though in the U.S., the Individuals with Disabilities Education Act (2004) does not mandate transition planning before the age of 16 [74]. There is a global need for policies that create the conditions for self-determination and the use of effective and culturally valid self-determination practices.

## 5. Best Practices to Measure and Promote Self-Determination

Researchers have explored tools to assess students’ self-determination and provide corresponding instruction and supports, and below we describe the research base for specific measures and practices aligned with Causal Agency Theory [2].

### 5.1. Assessment

#### The Self-Determination Inventory

Utilizing valid and reliable assessment tools is essential to effectively promote student self-determination. This can inform the design of instructional needs to individualize supports and interventions as well as establish effective practices by tracking the outcomes of self-determination interventions. Researchers developed the Self-Determination Inventory System [SDIS; 13], a set of measures to collect self-determination data from youth and adults, which includes the Self-Determination Inventory: Student Report (SDI:SR), Self-Determination Inventory: Parent and Teacher Report (SDI: PTR), and Self-Determination Inventory: Adult Report (SDI:AR). There has been a focus on including students with and without disabilities in this work. This is because self-determination is relevant to all students, and there is a critical need for universal self-determination instruction in inclusive settings for students with and without disabilities, inclusive of students with intellectual and developmental disabilities.

The SDI:SR includes 21 items with the anchors of “Disagree” and “Agree”, ranging from 0 (low) to 99 (high), which are computer-scored. Scores are calculated by averaging all completed items (arithmetic mean), resulting in an overall self-determination score and subscale scores for DECIDE, ACT, BELIEVE. Students complete the SDI:SR online, and it has accessibility features, including in-text definitions and the use of a slider scale. The SDI:SR was validated with over 4000 secondary students with and without disabilities, inclusive of students from racially and ethnically marginalized backgrounds with different disability labels. This research has suggested that SDI:SR scores varied across racial/ethnic groups and disability categories and demonstrated a satisfactory internal consistency. Researchers have suggested that a general single factor representing overall self-determination best fits SDI:SR scores. However, more research is needed to ensure that measures of self-determination are accessible for students with intellectual and developmental disabilities, including students with complex support needs. This is particularly important as the existing validated measures of self-determination require students to perform reading and writing tasks which may create cognitive challenges for some students with intellectual and developmental disabilities. These measures also have been shown to have cross-cultural validity, with translations into multiple languages and cultures, as described earlier.

### 5.2. Intervention

#### 5.2.1. The Self-Determined Learning Model of Instruction

As introduced previously, the Self-Determined Learning Model of Instruction (SDLMI) is an evidence-based practice designed to enhance self-determination and support students in setting and pursuing their goals in secondary school [74]. The SDLMI is a model of instruction that enables trained implementers to teach skills associated with self-determination across settings, including transition-planning, academic instruction, and community-based settings. The SDLMI can therefore be implemented in many ways, such as one-on-one, small groups, and whole classes. The SDLMI can also be facilitated both in-person and virtually. The SDLMI includes three core elements, Student Questions, Teacher Objectives, and Educational Support, across three instructional phases, Phase 1: Set a Goal, Phase 2: Take Action, and Phase 3: Adjust Goal or Plan. Within each phase, students work through four Student Questions that engage them in self-directing the process of goal setting, action planning, and evaluating goal achievement. Through Teacher Objectives, implementers support students in addressing the four Student Questions (totaling 12 Student Questions) within each phase and incorporate Educational Supports that can be individualized based on each student’s strengths, assets, values, and self-identified areas of growth to teach self-determination skills. Multiple review studies have identified the SDLMI as a widely used self-determination intervention for students with intellectual and developmental disabilities and documented its impacts on student self-determination and goal-attainment outcomes (e.g., [67]). Yet only a limited number of studies have directly investigated the implementation of the SDLMI with racially and ethnically marginalized students with and without disabilities, an area for future research. While initially developed and evaluated in the U.S., the SDLMI has been adapted in other cultures, and research should continue to explore the cross-cultural validity [75,76].

#### 5.2.2. Decisional Supports

An increasing number of states and jurisdictions are adopting supported decision-making structures, legal arrangements which allow people with disabilities to make decisions about their lives with the support of trusted parties such as friends or family [25]. This aligns with the focus on legal agency in the UNCRPD. These legal structures serve as alternatives to substituted decision-making arrangements such as guardianships and conservatorships, through which third parties have legal authority to make decisions for disabled people.

## 6. Future Directions

While a substantial research base supports not only the importance of self-determination for transition-age students with intellectual and developmental disabilities but also validates the assessment measures and instructional framework, work remains to ensure equitable opportunities to develop and exercise self-determination are available to all across the world.

### 6.1. Cultural Validity

The development of self-determination is impacted by the cultural knowledge and values of students’ families and communities. For example, researchers have found a discrepancy between how families and teachers define and provide support and opportunities to promote self-determination at home or in the community, impacting self-determination [48]. Segregated settings and limited opportunities and support to build self-determination and other school outcomes are other systemic barriers that students with intellectual and developmental disabilities, including racially and ethnically marginalized youth with disabilities, encounter in schools [77,78]. These barriers reflect deficit-based approaches rooted in racism and ableism. Shawn Aleong, student researcher and self-advocate, confirmed this sentiment when he recalled his experiences trying to obtain supports. He felt that as an African American man and a man with a disability in the U.S., he found it twice as hard to receive the support he needed because of the systematic oppression of African Americans and people with disabilities. Aleong adds that, as a result, he feels there are two justice systems in the U.S.: one for people who are white and another for minority groups. Given the importance of inclusive environments that celebrate and elevate marginalized students’ cultural identities [79], more work is needed to advance culturally responsive teaching practices and support in self-determination interventions. Such efforts will advance the recognition of racially and ethnically marginalized youth with disabilities and capitalize on the funds of knowledge they bring to their school communities across the world [80].

### 6.2. Further Integration of Strengths- and Rights-Based Approaches

In describing the importance of both strengths- and rights-based approaches to promoting self-determination with transition-age youth with intellectual and developmental disabilities, it is critical to discuss the complexity of integrating these two approaches. Person-centered approaches place the strengths, goals, and preferences of the person at the center of the process, with consideration for the demands of the environment and the person’s support needs [51]. The underlying premise of rights-based approaches is that people with disabilities have the right to be treated equally to people without disabilities [81]. Glicksman and colleagues [82] propose a dialectical model to address the potential challenges with applying the two approaches simultaneously, with the goal of fully recognizing and upholding the rights of people with disabilities to self-determination while supporting them to work toward their person-centered goals. Glicksman et al. outline how rights-based approaches may fail to meet the support needs of people with disabilities and may, unintentionally, contribute to the lack of fulfillment of their person-centered vision. To add the perspective of a person with lived experience of disability, Shawn Aleong presents the counterargument that “able-bodied people sometimes focus on the support side and forget about the choice”. Aleong shared the story of his experiences with a support professional in the community. The support professional would make comments like, “I know what’s best for you”. The support professional may have believed their guidance on behavior in the community was essential for Aleong to achieve his person-centered goals, but in denying him choice, the person did not operate from a rights-based approach.

In the dialectical model from Glicksman et al. [82], support professionals and others working with people with disabilities must engage in an ongoing critical inquiry process to balance the respect for the rights of individuals to self-direct their actions and addressing support needs related to a mismatch between expectations in the environment and the skills and abilities of the person. Deeper theoretical work, too, is needed, to evaluate the distinction between social conventions and personal decisions, the distinction between informed decision making and impulsive behavior and how related needs are addressed within a supported decision-making model, and lastly, the danger of ableism influencing perspectives of support professionals even within the dialectical model. Considerations when addressing the integration of rights-based and strengths-based approaches to promote self-determination include the need for the training of support professionals and others on the dialectical model and self-determination and the potential inequity of resources, compensation, and personal factors (race, ethnicity, gender) across support professionals.

### 6.3. Systemic Changes in Schools, Communities, and Society

To fully support transition-age youth with intellectual and developmental disabilities in their pursuit of self-determined lives, systemic changes are needed in the places where they live, learn, and work. As Shawn Aleong, self-advocate, states about policies and practices that promote systemic change, there is a need to “learn from the source”. In other words, people with lived experience of disability offer the most valuable insight on barriers to their goal setting, problem solving, choice making, and other skills and abilities associated with self-determination. Understanding the lived experience of people with intellectual and developmental disabilities who are multiply marginalized because of racism, sexism, transphobia, xenophobia, and other forms of bias is even more critical. Transition-age youth of color report hurdles to engaging in self-determined action during their move from high school to the adult world rooted in racism and ableism [49]. Ongoing work is needed in research and practice to understand and change the systemic factors limiting students with intellectual and developmental disabilities who are multiply marginalized, particularly as research has documented such findings for decades.

Policy offers an important avenue to create systemic change, given its broad reach to students, adults, families, and professionals in the field. Recommendations from the research and practice described here offer insights into the system-wide change to promote self-determination more broadly and equitably, such as policy changes for how transition-age students and families receive information on supported-decision-making models and other alternatives to guardianships [83] and the integration of inclusive research as a standard for high-quality research [30]. The field of disability has made meaningful progress in expanding understandings of self-determination, developing its application to practice, and advocating for policy to protect the rights of people with disabilities to set and go after goals that are personally meaningful to them. However, as we look forward, it is essential to consider how the history of self-determination in research, policy, and practice and its current status offer direction across the world. Significant change remains needed, but work on critical topics such as inclusive research, multi-tiered systems of support, cultural responsivity, and measurement and practice development in line with the modern theory on self-determination offers a promising start to such change to build more inclusive, universal supports for personal and collective self-determination.

## Data Availability

Not applicable.

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
