# Peer review of "Self-Determination Research: Current and Future Directions"

_behavsci, 2024, doi:10.3390/bs14070613_

Round 1

Reviewer 1 Report

Comments and Suggestions for Authors

The manuscript is thoughtfully laid out and makes a great addition to the field. As a minor note, the abstract is not laid out in the order in which information is presented in the manuscript. For example, the research on self-determination is presented prior to the discussion of causal agency theory. Section 4 of the manuscript could be more thoughtfully described. Additionally, it is unclear if the tools described in Section 5 are truly derived from causal agency theory as suggested in the abstract, or if they are just 
"aligned with causal agency theory" as described in the narrative.

The final sentence of Section 1 of the manuscript could also more carefully lay out the organization of the paper as the transition from Section 3 to Section 4 is a bit hard to follow as both include theoretical models/frameworks.

Author Response

Comment 1: As a minor note, the abstract is not laid out in the order in which information is presented in the manuscript. For example, the research on self-determination is presented prior to the discussion of causal agency theory. Section 4 of the manuscript could be more thoughtfully described. Additionally, it is unclear if the tools described in Section 5 are truly derived from causal agency theory as suggested in the abstract, or if they are just "aligned with causal agency theory" as described in the narrative.

Response 1: Thank you for this feedback. We have edited the lay-out of content in the abstract to reflect the order of material in the manuscript. We have added a more robust description of Section 4 to the abstract, and we have edited the wording of Causal Agency Theory as aligned with the assessment and intervention described.

Comment 2: The final sentence of Section 1 of the manuscript could also more carefully lay out the organization of the paper as the transition from Section 3 to Section 4 is a bit hard to follow as both include theoretical models/frameworks.

Response 2: We appreciate this feedback and revised the final sentence of Section 1 to reflect that Section 3 broadly discusses the status of knowledge, practice, and policy on self-determination, and Section 4 specifically addresses meeting the support needs of transition-age students with intellectual and developmental disabilities.

Reviewer 2 Report

Comments and Suggestions for Authors

Dear authors,
The article aims to investigate the current state of self-determination of students with intellectual disabilities and their transition from school to adult life, among other aspects.
The study is interesting, and gives a great contribution to current research on the subject.
The objective of the study has been correctly specified in the abstract.
The authors clearly situate the current state of research on self-determination in the world.
Perhaps it would be interesting if, when commenting that operationalisation varies according to cultures and cultural identities, they could specify some examples.
More research should be done on the social and community factors that are linked to the issue.
It would also be interesting to cite more studies at the international level (other countries as they see it) to see what the literature says about this aspect and the high pressures these people and their families are under in order to achieve self-determination.

This is a frankly important issue as it highlights a serious problem in society at large.
It would be interesting to highlight what mechanisms other countries have in place to address the issue. This is very important in this article, as it seems that there is a lack of data in the theoretical framework to be able to go into more detail.
One could inquire further into why the United States remains one of the few countries that until these last decades has not developed a policy that could be seen as fostering the spirit of self-determination.
I agree with the authors that further research is needed on the alignment of disability policy with the UNCRPD and its emphasis on self-determination.
I have found it very interesting that the authors work with inclusive research, in which people from the communities of interest participate as members of the research team. This type of research allows the research team to gain the perspective of people with lived experience.

One point to note is the lack of information on the demographics of the participants in these studies.

It would be advisable to elaborate further on the barriers and obstacles faced by persons with disabilities in exercising self-determination, such as discrimination, lack of accessibility and lack of adequate support. It would also be interesting to include concrete examples of programmes or initiatives that have been successful in promoting the self-determination of persons with disabilities.
Overall, this article offers a contribution to the debate on equal rights for persons with disabilities and the importance of promoting their self-determination as a fundamental part of their dignity and autonomy.

However, the authors do not mention whether specific criteria were used to include or exclude certain studies, which could affect the objectivity and completeness of the literature review.
It would be useful to explain whether the date of publication, the relevance of the study, the methodology used, among other factors, were taken into account in determining which literature sources were included in the analysis.
The authors have not developed the discussion and conclusions.
Nor do I see that they have described the limitations of the research.
The study is interesting because it can help to establish appropriate plans, policies and strategies for improvement. However, I am concerned that no results are shown.
The questions that have guided this study should be clarified beforehand.
The manuscript has some difficulties in drafting, and although it is relevant, it should be presented in a better structured way throughout the text.
The total number of articles potentially relevant to the study is sufficient.
It is unclear to me what inclusion/exclusion criteria the authors have taken into account in order to select the bibliography on which they have been based.
The work is adapted to the scope of the journal.
As I have said, the article could be improved with the comments I have made previously.
I consider accepting the manuscript in its present form with the following minor suggestions I comment.

Kind regards

Author Response

Comment 1: The authors clearly situate the current state of research on self-determination in the world. Perhaps it would be interesting if, when commenting that operationalisation varies according to cultures and cultural identities, they could specify some examples. More research should be done on the social and community factors that are linked to the issue. It would also be interesting to cite more studies at the international level (other countries as they see it) to see what the literature says about this aspect and the high pressures these people and their families are under in order to achieve self-determination. This is a frankly important issue as it highlights a serious problem in society at large. It would be interesting to highlight what mechanisms other countries have in place to address the issue. This is very important in this article, as it seems that there is a lack of data in the theoretical framework to be able to go into more detail. One could inquire further into why the United States remains one of the few countries that until these last decades has not developed a policy that could be seen as fostering the spirit of self-determination.

Response 1: We agree with the important points noted above. We added an example from a study in Turkey detailing how self-determination is operationalized uniquely based on culture (p. 2, lines 61-65). Additionally, we have reduced the research cited from the United States to allow for a greater focus on self-determination internationally.

Comment 2: One point to note is the lack of information on the demographics of the participants in these studies.

Response 2: Yes, we agree with the importance and named this issue on p 3, lines 155-157.

Comment 3: It would be advisable to elaborate further on the barriers and obstacles faced by persons with disabilities in exercising self-determination, such as discrimination, lack of accessibility and lack of adequate support. It would also be interesting to include concrete examples of programmes or initiatives that have been successful in promoting the self-determination of persons with disabilities.

Response 3: We appreciate this important point and have added research on barriers, while also noting the need for this work to extend to transition-age youth with intellectual and developmental disabilities across international settings (see p. 4, lines 163-170).

Comment 4: Overall, this article offers a contribution to the debate on equal rights for persons with disabilities and the importance of promoting their self-determination as a fundamental part of their dignity and autonomy. However, the authors do not mention whether specific criteria were used to include or exclude certain studies, which could affect the objectivity and completeness of the literature review. It would be useful to explain whether the date of publication, the relevance of the study, the methodology used, among other factors, were taken into account in determining which literature sources were included in the analysis. The authors have not developed the discussion and conclusions. Nor do I see that they have described the limitations of the research. The study is interesting because it can help to establish appropriate plans, policies and strategies for improvement. However, I am concerned that no results are shown. The questions that have guided this study should be clarified beforehand. The manuscript has some difficulties in drafting, and although it is relevant, it should be presented in a better structured way throughout the text. The total number of articles potentially relevant to the study is sufficient. It is unclear to me what inclusion/exclusion criteria the authors have taken into account in order to select the bibliography on which they have been based.

Response 4: We appreciate this feedback and would note that because this was not a systematic review of the literature, we did not have specific research questions, methods, or results. We have made revisions to the framing of the abstract to clarify the type of manuscript and the approach.